# Associations between area-level socioeconomic disadvantage and COVID-19 disease consequences in Sydney, Australia: A retrospective cohort analysis

**Christopher Standen**[1]¤*, **Esther Tordjmann**[2]¤, **James Wood**[2]¤, **Fiona Haigh**[1]¤

**1** International Centre for Future Health Systems, University of New South Wales, Sydney New South Wales, Australia, **2** School of Population Health, University of New South Wales, New South Wales, Australia

¤ Current address: University of New South Wales, Sydney, New South Wales, Australia
* c.standen@unsw.edu.au

## Abstract

### Background

Socioeconomic disparities have shaped COVID-19 outcomes worldwide. Focusing on disease consequences once infected (severity among cases), we examined whether area-level socioeconomic disadvantage was associated with hospitalisation and death among COVID-19 cases in Greater Sydney, Australia.

### Methods

We conducted a retrospective cohort study of confirmed and probable COVID-19 cases recorded in the New South Wales Notifiable Conditions Information Management System from 2 March 2020–21 February 2022. Area-level disadvantage was measured using the Index of Relative Socio Economic Disadvantage (IRSD). We modelled the odds of (a) hospitalisation and (b) death conditional on infection using logistic regression, adjusting for age, gender and pandemic phase.

### Results

Among 782,883 included cases, 3.5% were hospitalised and 0.2% died due to COVID-19. After adjustment, a 100-point increase in IRSD score (indicating less socioeconomic disadvantage) was significantly associated with lower odds of hospitalisation (adjusted odds ratio [AOR] 0.80, 95% CI 0.79–0.81) and death (AOR 0.78, 95% CI 0.74–0.83). There was no evidence that these associations differed across pandemic phases.

### Conclusions

Area-level socioeconomic disadvantage was associated with higher risks of hospitalisation and death among COVID-19 cases in Greater Sydney – a setting with public

**Data availability statement:** The data underlying this study are third-party public health surveillance records held by NSW Health and cannot be publicly shared due to legal and ethical restrictions under NSW Health data governance and privacy policies. Qualified researchers may request access to NCIMS data through NSW Health's data governance processes (contact: moh-datagovernance@health.nsw.gov.au or the NSW Health data-sharing portal). Access is granted to researchers who meet criteria for confidential data access and have approved ethics and governance documentation.

**Funding:** This work was funded by Sydney Local Health District through a memorandum of understanding with the University of New South Wales.

**Competing interests:** The authors have declared that no competing interests exist.

hospital care – with some postal areas having more than twice the odds of hospitalisation and death as others. Given the absence of linked comorbidity and vaccination data, the most plausible explanation is disparities in comorbidity and risk-factor burden, although contributions from differences in access to testing and quality of care cannot be ruled out. Public health responses should prioritise chronic disease prevention and management in disadvantaged communities to mitigate inequitable outcomes in future pandemics.

## Background

Disparities in COVID-19 disease outcomes have highlighted the role of socioeconomic disadvantage. While individual risk factors such as older age, sex, and smoking are consistently associated with increased risks of hospitalisation and death [1–3], international evidence shows these risks are amplified among socioeconomically disadvantaged populations. People living in lower-income or deprived areas have experienced higher hospitalisation and mortality risk in the United States, the United Kingdom, and mainland Europe [4,5].

Katikireddi et al. [6] outline six pathways through which socioeconomic disadvantage can lead to unequal pandemic outcomes, spanning differential exposure, vulnerability, disease consequences, and social and policy responses. Our analysis targets the pathway most directly relevant to clinical outcomes among confirmed cases: differential disease consequences once infected. By modelling hospitalisation and death among notified COVID-19 cases, we examine inequalities in severity across socioeconomic strata, conditional on infection, rather than inequalities arising from exposure or susceptibility. Other pathways in the framework, including differential social consequences and effects of control measures, are therefore outside the analytical scope of this study.

Most prior studies have focused on disparities in absolute risks of hospitalisation and death. Fewer have examined outcomes conditional on infection – i.e., differential disease consequences (Katikireddi et al.'s third pathway) – which may reflect differences in comorbidity burden and vaccination status, as well as access to and quality of treatment [7].

The Australian context offers an opportunity to investigate differential disease consequences across socioeconomic strata within a health system that provides public hospital care. A cohort study in Western Sydney Local Health District (WSLHD), covering a population of approximately 1.1 million, reported no statistically significant association between area-level socioeconomic disadvantage and a composite measure of severe outcome (death, intensive care unit (ICU) admission, mechanical ventilation) after adjustment for age, gender, number of comorbidities, and vaccination status – attributing this finding to equitable access to high-quality care [8]. Meanwhile, a cohort study in Sydney Local Health District (SLHD), covering a population of approximately 740,000, reported statistically significant associations between area-level socioeconomic disadvantage and rates of hospitalisation and

death, adjusting for age and gender [7,9]. This finding was replicated in a study that focused on the period of public health restrictions (16 June 2021–18 February 2022) and also adjusted for vaccination status and SARS-CoV-2 variant [10].

This study extends the analysis to the entire Greater Sydney metropolitan area (>5 million residents), which has substantial socioeconomic heterogeneity. Guided by Katikireddi et al.'s framework, we focus on differential disease consequences, examining whether, among reported COVID-19 cases, area-level socioeconomic disadvantage is associated with (a) hospitalisation risk and (b) fatality risk, after controlling for age, gender and pandemic phase. By moving from a single-district to a metropolitan scale, adjusting for pandemic phase, and disaggregating adverse outcomes, we aim to provide a clearer picture of equity in COVID-19 outcomes in the context of a health system providing public hospital care.

## Methods

### Study design and setting

We conducted a retrospective cohort study to investigate the association between area-level socioeconomic disadvantage and COVID-19 hospitalisation and fatality risks among confirmed and probable cases using routinely collected surveillance data. The study was conducted in Greater Sydney in the state of New South Wales (NSW), Australia's most populous metropolitan region, with an estimated population of 5.2 million in 2021. Area-level socioeconomic disadvantage, as measured using the Index of Relative Socio-Economic Disadvantage (IRSD), varies substantially across the region (Fig 1).

The study period commenced on 2 March 2020, when the first locally acquired case of COVID-19 in Greater Sydney was reported, and ended on 21 February 2022, by which time emergency public health restrictions had ended.

Vaccination for COVID-19 in Australia commenced in February 2021. Coverage was still relatively low at the outset of the Delta pandemic phase: at 29 June 2021, 830,884 doses had been administered among NSW's 6.6 million residents aged ≥16 years [11]. By the end of the Delta pandemic phase (30 November 2021), 92.4% of the eligible population aged ≥16 years in NSW had received two doses of a COVID-19 vaccine. Coverage increased further during the Omicron phase, reaching 94.3% two-dose coverage by the end of our study period (21 February 2022) [12].

In Australia, most primary care, specialist and diagnostic services are delivered through the private sector, with the federal government rebating some or all the cost to Medicare card holders (Australian citizens, permanent residents, and eligible visitors). However, throughout the study period, polymerase chain reaction (PCR) testing for COVID-19 was provided without out-of-pocket cost through both public services and private pathology providers. From 12 January 2022, self-reporting of positive COVID-19 rapid antigen tests (RATs) was mandatory and could be done via the 'Service NSW' app or website, or by phone.

Hospitalisations captured in this study occurred predominantly in public hospitals, with some admissions managed in private hospitals under arrangements in place during the public health emergency. Hospital care for COVID-19 was provided free to all Medicare card holders. Nevertheless, barriers to accessing testing and healthcare may have persisted, particularly in socioeconomically disadvantaged communities [7,13].

This study formed part of a broader, equity-focused health impact assessment (EFHIA) of the COVID-19 pandemic and associated public health responses in Sydney, Australia [7,9].

Ethics approval for the study was granted by the Sydney Local Health District Research Ethics and Governance Office on 11 December 2020 (Protocol No. X20-0467 and 2020/ETH02564). Informed consent was not required for retrospective analysis of de-identified surveillance data.

### Data sources

COVID-19 case data were obtained from the Notifiable Conditions Information Management System (NCIMS), maintained by the NSW Ministry of Health, on 22 February 2022. NCIMS data were de-identified before analysis by the corresponding/lead author (CS). Other authors did not have access to information that could identify individual cases during or after

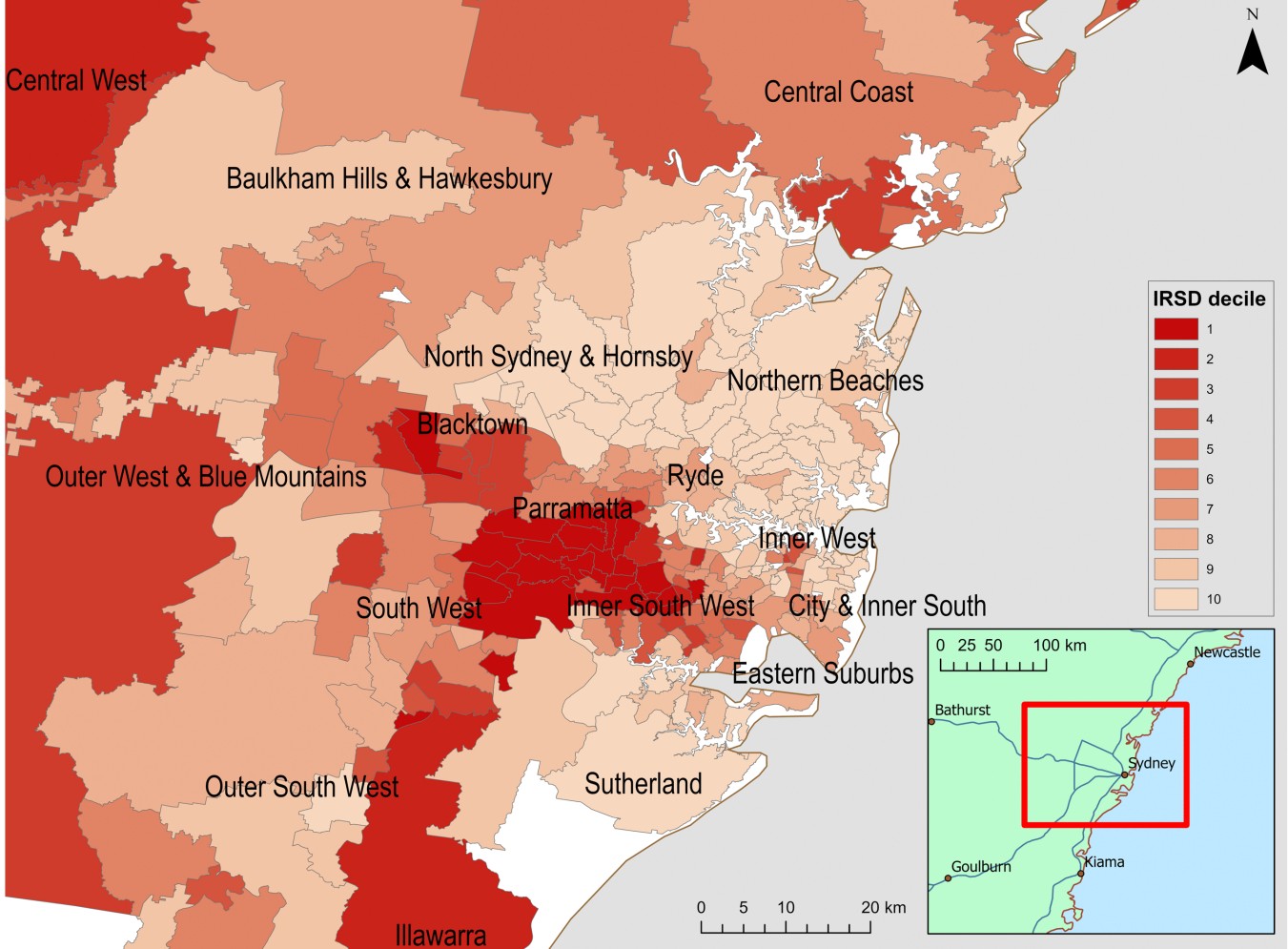

**Fig 1. Map of Greater Sydney showing Index of Relative Socio-Economic Disadvantage (IRSD) deciles for postal areas.** Basemap source: Natural Earth (public domain).

data collection. Demographic and socioeconomic data were sourced from the 2021 Census of Population and Housing [14] and the Socio-Economic Indexes for Areas (SEIFA) [15], both published by the Australian Bureau of Statistics.

## Variables

The primary outcomes were hospitalisation and death due to COVID-19, recorded as dichotomous variables in the NCIMS database. The key independent variable was area-level socioeconomic disadvantage, specifically, the IRSD score corresponding to each case's residential postal area. An IRSD score for a postal area is a weighted composite measure of 16 economic and social disadvantage indicators for that postal area, including the percentage of people aged 15 years and over who have no educational attainment, and the percentage of people who are unemployed [16]. IRSD scores are standardised to have a mean of 1,000 and a standard deviation of 100 across Australia. In the study area, IRSD scores ranged from 711 to 1,122, with a mean of 1,029 and a standard deviation of 78. IRSD scores were linked algorithmically to NCIMS records and, to aid interpretation of effect estimates, rescaled and modelled per 100-point increase, with higher values indicating less disadvantage.

Covariates from NCIMS records included age group (categorical), gender (categorical), and pandemic phase – the latter derived from diagnosis date and categorised to capture major temporal changes in dominant variants, control measures, and population immunity. Pandemic phase was categorised as Ancestral (2 March 2020–25 June 2021), Delta (26 June to 30 November 2021), and Omicron (1 December 2021–21 February 2022).

## Analysis

Initial data cleaning involved the exclusion of cases acquired overseas. Descriptive analysis compared distributions of cases, hospitalisations and fatalities across gender, Indigenous status, and age groups with those of the Greater Sydney population [14]. Descriptive spatial analyses were conducted using ArcGIS software [17] to map incidence of COVID-19 cases, case hospitalisation rate, and case fatality rate. An epidemic curve was constructed by aggregating notified COVID-19 cases by date of symptom onset and plotting daily case counts with a 7-day rolling average. Further exploratory analysis involved examining relationships between area-level socioeconomic disadvantage (IRSD score) and COVID-19 cases, hospitalisations and fatalities using scatterplots and Pearson correlation tests. These analyses were conducted in Python [18].

Multivariate logistic regression models were then applied to assess the associations between IRSD score and two key outcomes: the risk of hospitalisation and the risk of death due to COVID-19. In both models, covariates included age, gender and pandemic phase.

To assess whether associations between socioeconomic disadvantage and outcomes varied over time, models including an interaction term between IRSD and pandemic phase were fitted. As these interaction terms were not statistically significant for either outcome, final models are presented without them.

The Box-Tidwell procedure was used to confirm a linear relationship between IRSD score and the logit transformation of each dependent variable. Variance inflation factors were calculated to confirm the absence of multicollinearity among the independent variables. Regression modelling was carried out using IBM SPSS Statistics [19]. We followed STROBE reporting guidelines for observational studies.

## Results

Fig 2 shows the epidemic curve of notified COVID-19 cases in Greater Sydney over the study period. Case counts varied markedly across time, with relatively low case numbers during the Ancestral phase, a substantial wave during the Delta phase, and a large surge during the Omicron phase.

After excluding 2,527 cases acquired overseas, there were 782,883 reported confirmed or probable COVID-19 cases in the study area during the study period, with 121,828 (15.6%) being self-reported RAT results. Of these cases, 27,122 (3.5%) were hospitalised, 2,280 (0.3%) were admitted to an ICU, and 1,491 (0.2%) were recorded as dying due to COVID-19. The distribution of these cases across gender, age and Indigenous status is presented in Table 1. The proportion of cases in the 60+ age group (10.2%) was half the corresponding population proportion (20.5%).

Figs 3–5 show the spatial distributions of cases per 100,000 persons, hospitalisations per 100,000 cases, and fatalities per 100,000 cases, respectively.

The scatterplots (Fig 6) indicated a linear relationship between IRSD score and COVID-19 outcomes. At the postal area level, there was a strong negative correlation between IRSD score and both case incidence (Fig 6) and hospitalisation risk (Fig 7). There was a weak negative correlation between IRSD score and fatality risk (Fig 8).

For the logistic regression analyses of hospitalisation risk and fatality risk, cases with incomplete/unknown information, or belonging to a category with a very low frequency were dropped, leaving 779,774 cases.

The logistic regression model of case hospitalisation risk (Table 2) was statistically significant ($\chi^2(7) = 34{,}591.87$, $p < .001$). The model explained 16.7% (Nagelkerke $R^2$) of the variance in hospitalisation risk and correctly classified 96.6% of cases. A 100-point increase in IRSD score (i.e., decreasing socioeconomic disadvantage) was

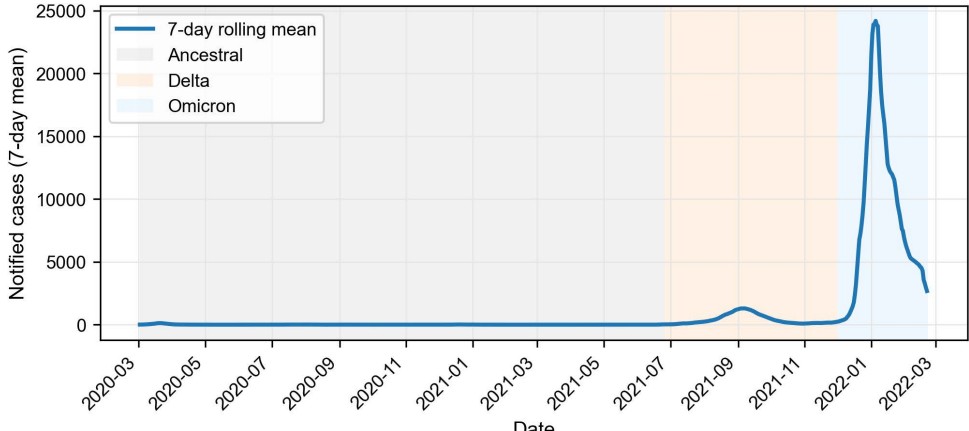

**Fig 2. Epidemic curve of notified COVID-19 cases in Greater Sydney, 2 March 2020 to 21 February 2022.**

**Table 1. Characteristics of reported COVID-19 cases, hospitalisations and fatalities in Greater Sydney (to 21 February 2022).**

| Characteristic | Number of reported COVID-19 cases(N = 782,883) | Number of reported COVID-19 hospitalisations (N = 27,122) | Number of reported COVID-19 fatalities (N = 1,491) | Greater Sydney population 2021(N = 5,231,147) |
|---|---|---|---|---|
| Gender/sex[a] | | | | |
| Female | 393,253 (50.2%) | 14,072 (51.9%) | 580 (38.9%) | 50.6% |
| Male | 388,668 (49.6%) | 13,033 (48.1%) | 911 (61.1%) | 49.4% |
| Transgender | 2 (<0.1%) | 1 (<0.1%) | 0 (0.0%) | – |
| Not stated/inadequately described | 960 (0.1%) | 16 (<0.1%) | 0 (0.0%) | – |
| Indigenous status | | | | |
| Not Indigenous | 257,338 (32.9%) | 17,639 (65.0%) | 1,308 (87.7%) | 94% |
| Indigenous | 18,599 (2.4%) | 964 (3.6%) | 18 (1.2%) | 1.7% |
| Not stated/unknown | 506,946 (64.8%) | 8,519 (31.4%) | 165 (11.1%) | 4.3% |
| Age group | | | | |
| 0–4 | 38,353 (4.9%) | 1,243 (4.6%) | 1 (0.1%) | 6.0% |
| 5–19 | 156,759 (20.0%) | 1,894 (7.0%) | 0 (0.0%) | 18.1% |
| 20–39 | 325,116 (41.5%) | 7,916 (29.2%) | 26 (1.7%) | 30.0% |
| 40–59 | 182,877 (23.4%) | 6,502 (24.0%) | 127 (8.5%) | 25.5% |
| 60+ | 79,709 (10.2%) | 9,567 (35.3%) | 1,337 (89.7%) | 20.5% |
| Not stated/unknown | 69 (<0.1%) | 0 (0.0%) | 0 (0.0%) | – |

[a] For COVID-19 cases, hospitalisations and deaths, this table reports the gender distribution. For the Greater Sydney population, this table reports the sex distribution from the 2021 Census of Population and Housing. The NCIMS does not record sex and does not allow any gender categories other than the four listed to be recorded. The 2021 Census did not allow respondents to report their gender. It allowed respondents to select from three categories for the sex question: male, female, and non-binary sex. However, Census products report data from the sex question as male and female only.

associated with a decrease in the risk of hospitalisation (adjusted odds ratio (AOR) 0.80, 95% CI 0.79–0.81). Risk of hospitalisation increased with age, with a case aged 50–69 two times as likely to be hospitalised as a case aged 20–49 (AOR 2.04, 95% CI 1.97–2.11), and a case aged 70 + 10 times as likely to be hospitalised (AOR 10.02, 95% CI 9.68–10.38). The hospitalisation risk for men was about 11% less than that for women (AOR 0.89, 95% CI 0.87–0.91).

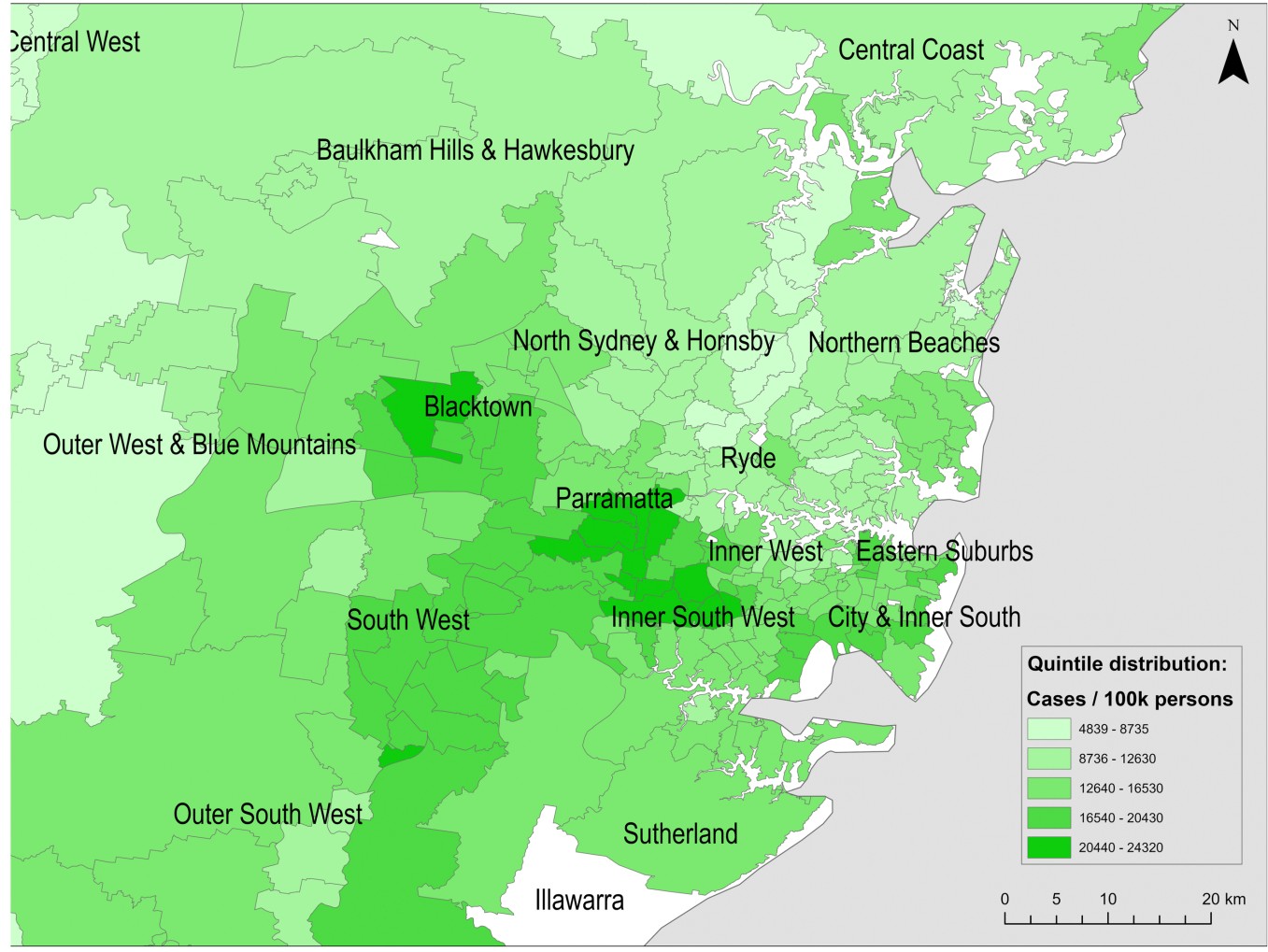

**Fig 3. Spatial distribution of reported COVID-19 cases per 100,000 persons across Greater Sydney postal areas (to 21 February 2022).** Base-map source: Natural Earth (public domain).

Similarly, the logistic regression model of case fatality risk (Table 2) was statistically significant ($\chi^2(7)$ = 7,594.58, $p<.001$). The model explained 35.4% (Nagelkerke $R^2$) of the variance in fatality risk and correctly classified 99.8% of cases. A 100-point Increase in IRSD score (i.e., decreasing socioeconomic disadvantage) was associated with a decrease in the risk of death due to COVID-19 (AOR 0.78, 95% CI 0.74–0.83). Risk of death due to COVID-19 increased with age: a case aged 50–69 was about 14 times as likely to die as a case aged 20–49 (AOR 14.16, 95% CI 10.80–18.57) and a case aged 70+was more than 200 times as likely to die due to COVID-19 (AOR 235.37, 95% CI 183.21–302.39). Male cases were more likely to die due to COVID-19 than female ones (AOR 1.56, 95% CI 1.40–1.74).

## Discussion

This study investigated associations between area-level socioeconomic disadvantage and disease consequences once infected with SARS-CoV-2 – the third pathway in Katikireddi et al.'s framework for understanding COVID-19 inequities [6] – in the Greater Sydney metropolitan region. After controlling for age, gender and pandemic phase, COVID-19 cases

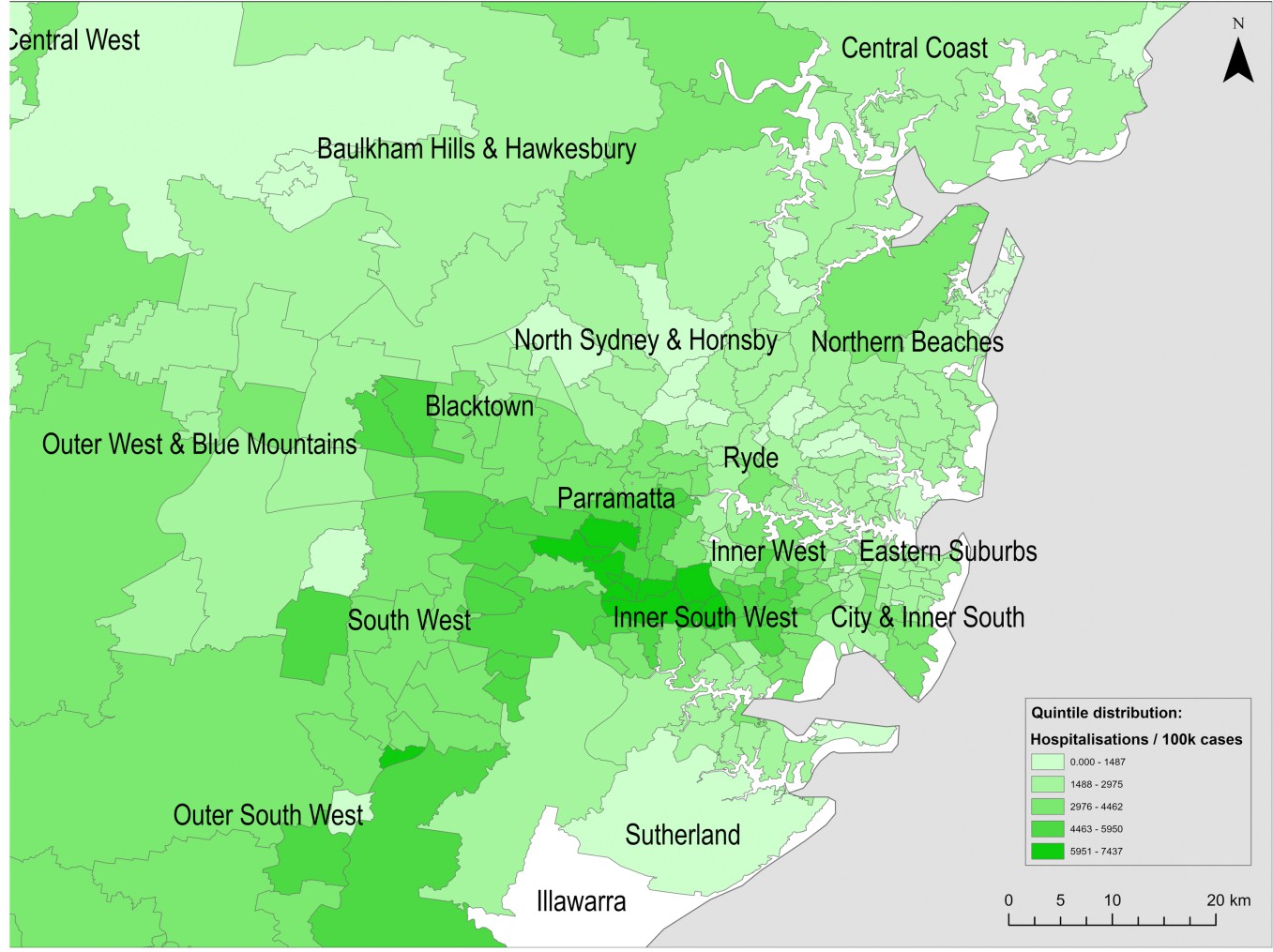

**Fig 4. Spatial distribution of reported COVID-19 hospitalisations per 100,000 cases across Greater Sydney postal areas (to 21 February 2022).** Basemap source: Natural Earth (public domain).

residing in more socioeconomically disadvantaged postal areas had higher odds of hospitalisation and of death due to COVID-19. To illustrate the magnitude of this gradient, the difference in area-level socioeconomic disadvantage (IRSD score) between Fairfield (postal area 2165; IRSD 758) and Northbridge (postal area 2063; IRSD 1,122) corresponds to approximately a 2.2-fold difference in the adjusted odds of hospitalisation and a 2.4-fold difference in the adjusted odds of death. We found no evidence that the strength of this association differed across Ancestral, Delta and Omicron pandemic phases, suggesting a stable association over time.

Male cases were less likely than female cases to be hospitalised but more likely to die due to COVID-19. This may reflect biological differences, as well as differences in thresholds for hospital admission; for example, pregnancy-related admissions among female cases.

The low number of deaths recorded among the 60 + age group relative to the population should be interpreted in the context of targeted infection control measures implemented in high-risk settings such as aged care facilities during the study period [7,20].

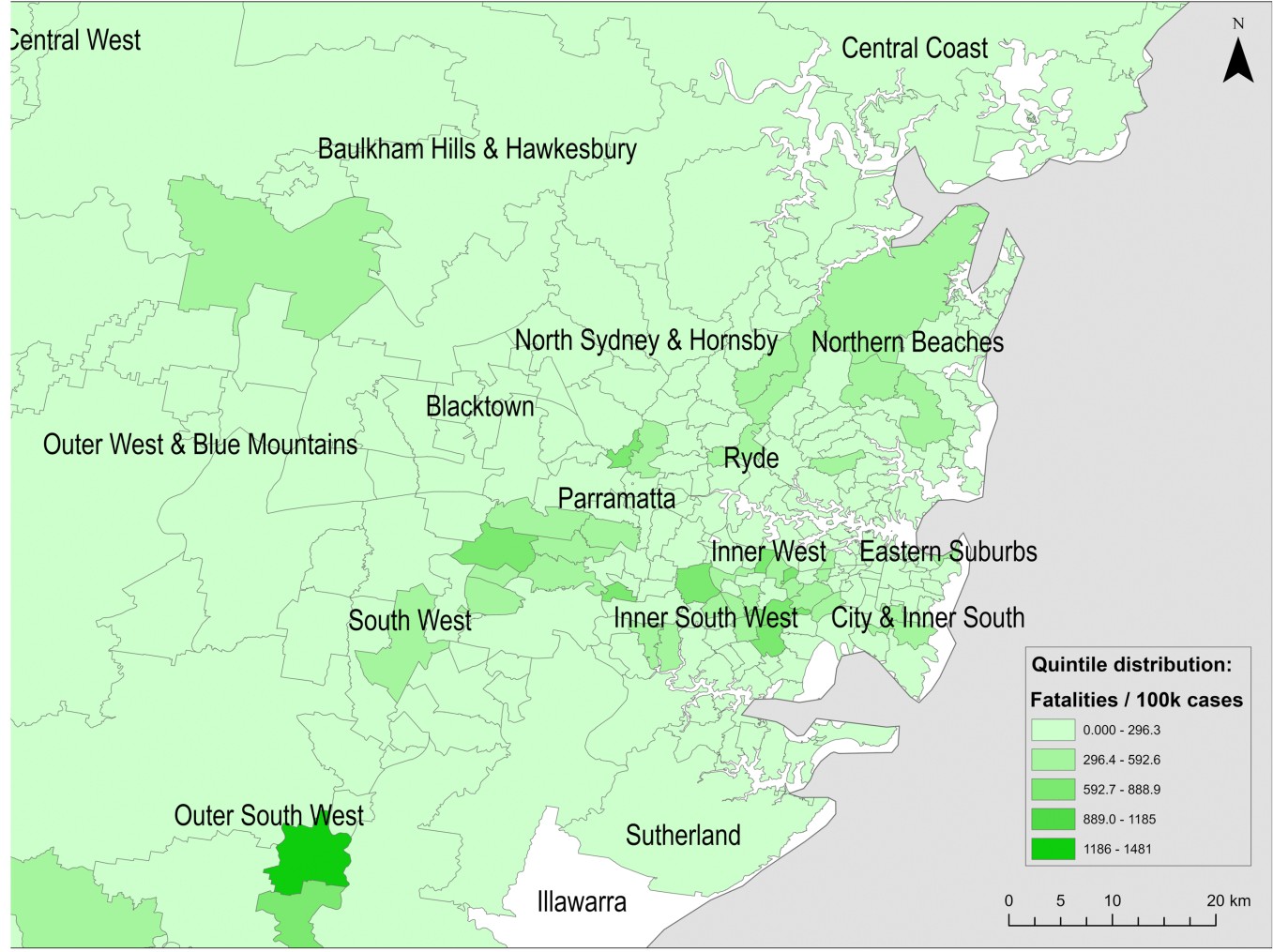

**Fig 5. Spatial distribution of reported COVID-19 fatalities per 100,000 cases across Greater Sydney postal areas (to 21 February 2022).** Base-map source: Natural Earth (public domain).

Our findings differ from a cohort analysis confined to Western Sydney Local Health District, which reported no association between area-level socioeconomic disadvantage and a composite measure of severe outcomes, after adjustment for age and comorbidities [8]. Differences in population size, socioeconomic heterogeneity, outcome definition, inclusion of deaths occurring outside hospital, model specification, and ability to adjust for comorbidities and vaccination status may account for the contrasting findings.

While our study design does not allow for the causes of inequalities in disease consequences to be determined, evidence from elsewhere suggests that disparities in comorbidity burden, smoking history, and in access to and quality of healthcare, may have played a role [1–3]. The WSLHD study's null association after comorbidity adjustment is consistent with this mechanism, though it did not report if severe outcomes were associated with socioeconomic disadvantage when the comorbidities covariate was omitted from the model.

Despite Australia's public hospital system, disparities in access to and quality of healthcare, cannot be ruled out as a contributory factor. The broader equity-focused health impact assessment (EFHIA) of the COVID-19 pandemic and

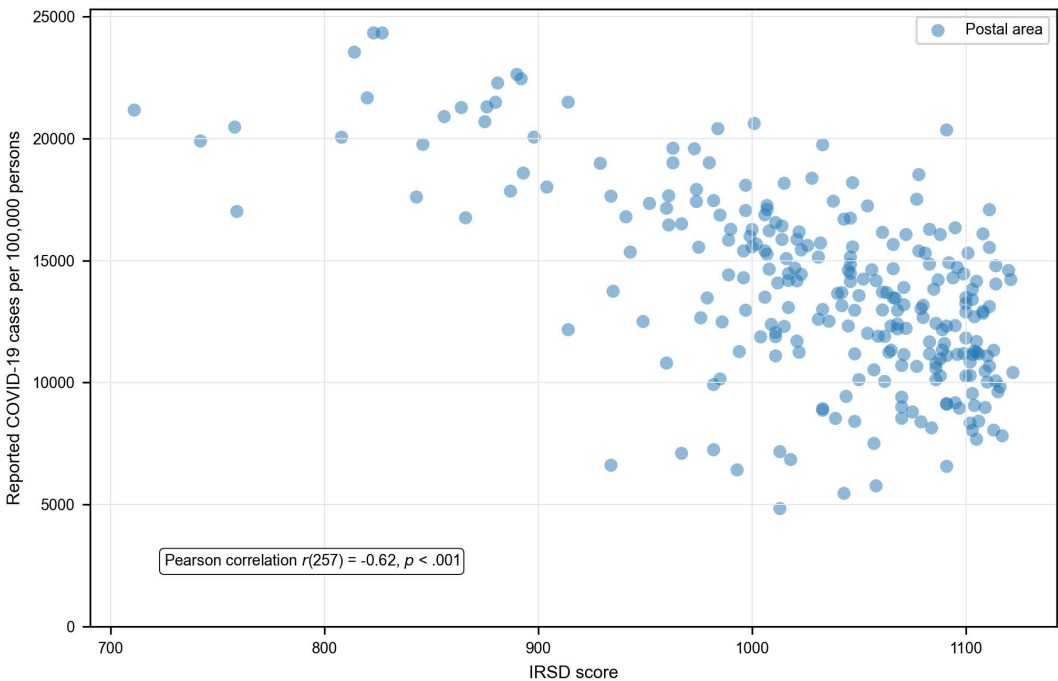

**Fig 6. Relationship between area-level socioeconomic disadvantage and reported COVID-19 case incidence.**

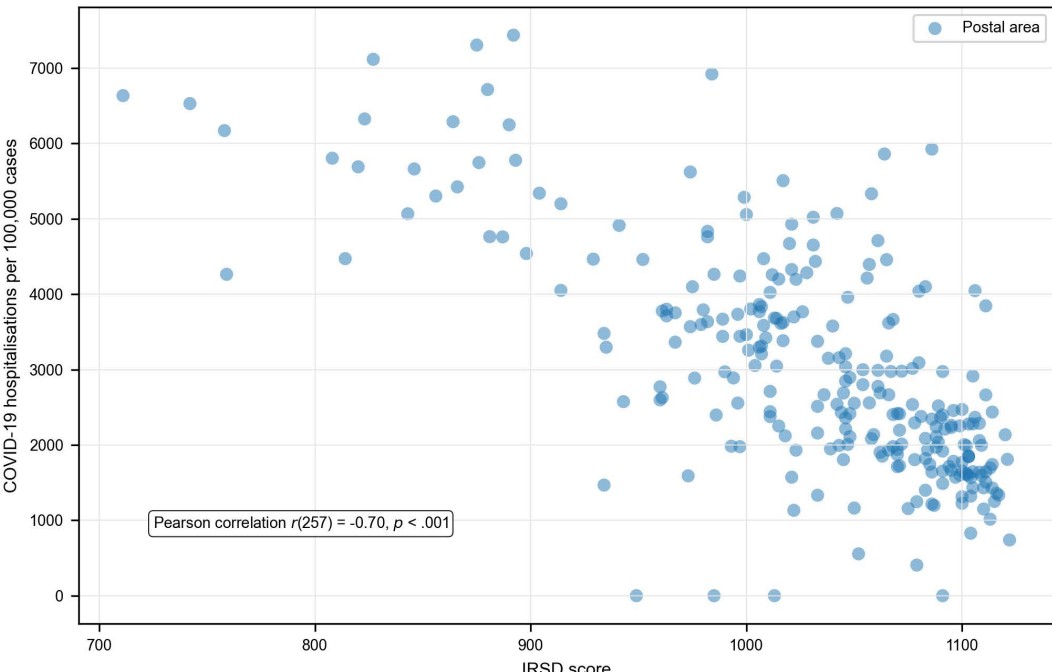

**Fig 7. Relationship between area-level socioeconomic disadvantage and COVID-19 case hospitalisation rate.**

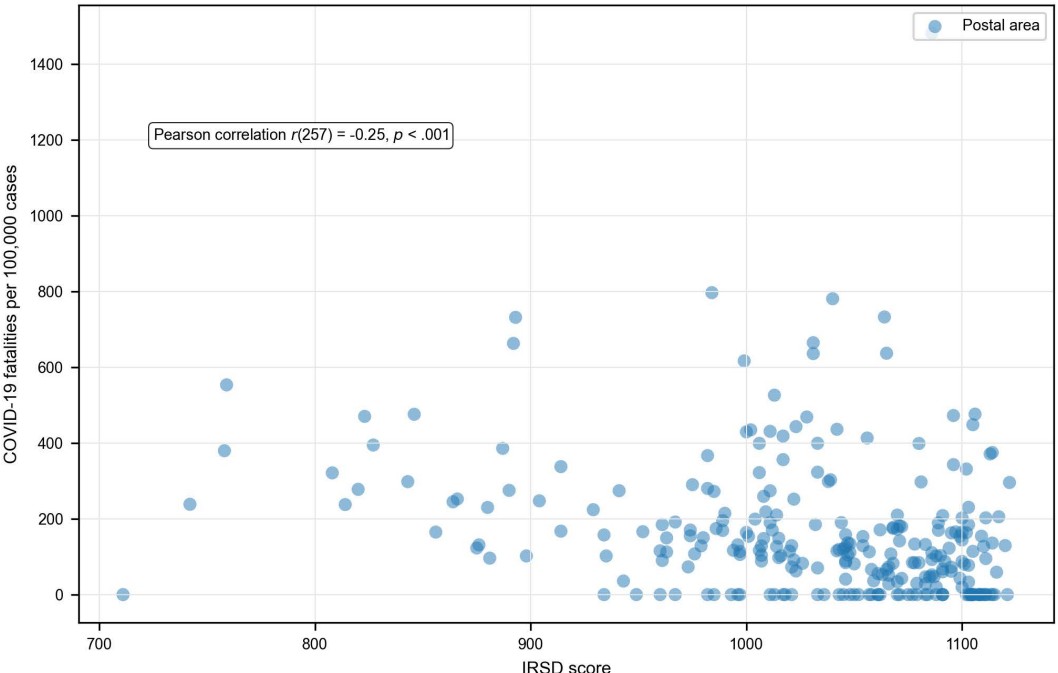

**Fig 8. Relationship between area-level socioeconomic disadvantage and COVID-19 case fatality rate.**

**Table 2. Logistic regression models of COVID-19 case hospitalisation and case fatality.**

| | Case hospitalisation | | | Case fatality | | |
|---|---|---|---|---|---|---|
| | AOR | 95% CI | *p* | AOR | 95% CI | *p* |
| Independent variables | | | | | | |
| *Socioeconomic disadvantage* | | | | | | |
| IRSD score (100-point increase) | 0.802 | 0.791–0.813 | < .001* | 0.784 | 0.744–0.827 | < .001* |
| *Gender* | | | | | | |
| Female[a] | 1.000 | | | | | |
| Male | 0.889 | 0.867–0.912 | < .001* | 1.560 | 1.402–1.737 | < .001* |
| *Age group* | | | | | | |
| 0–19 years | 0.525 | 0.504–0.546 | < .001* | 0.030 | 0.004–0.214 | < .001* |
| 20–49 years[a] | 1.000 | | | 1.000 | | |
| 50–69 years | 2.039 | 1.974 –2.106 | < .001* | 14.164 | 10.802–18.574 | < .001* |
| 70 + years | 10.024 | 9.681–10.378 | < .001* | 235.374 | 183.207–302.394 | < .001* |
| *Pandemic phase* | | | | | | |
| Ancestral[a] | 1.000 | | | | | |
| Delta | 1.705 | 1.470–1.978 | < .001* | 0.687 | 0.485–0.973 | .034 |
| Omicron | 0.246 | 0.212–0.285 | < .001* | 0.106 | 0.075–0.149 | < .001* |
| Model performance | | | | | | |
| Chi-squared test | *p*<.001 | | | *p*<.001 | | |
| Nagelkerke#39;s $R^2$ | .167 | | | .354 | | |

[a] Reference category. * Statistically significant at or above the 99.9% confidence level.

associated public health responses in Sydney [7,9] – of which this study forms one component – documented healthcare access barriers during the pandemic (e.g., longer wait times to see general practitioners in more disadvantaged areas, and exclusion from telehealth services for people without Internet access or with low digital literacy). The EFHIA also found that infection control measures (e.g., limits on gatherings) disrupted community networks, informal support systems and culturally appropriate community health services.

## Strengths and limitations

Strengths of this study include metropolitan-wide coverage, substantial socioeconomic heterogeneity, and the inclusion of deaths in non-hospital settings (e.g., aged care). The analysis explicitly targets disease consequences once infected. A further strength is the explicit adjustment for pandemic phase, which improved model fit and reduced temporal confounding arising from changes in dominant variants, infection control measures, and population immunity over the study period.

Key limitations include the use of an aggregate, area-level measure of socioeconomic disadvantage (IRSD) to infer the socioeconomic status of individual cases. Furthermore, the IRSD index is an imperfect measure of socioeconomic disadvantage and includes some potentially problematic variables [21], such as the percentage of households with no motor vehicle (City of Sydney, a local government area with a relatively low rate of household motor vehicle ownership, is one of the less disadvantaged in Greater Sydney).

We were unable to adjust directly for potentially important covariates, including vaccination status, comorbidities, and smoking history, due to limitations in the available data. This is reflected in the relatively low goodness-of-fit values in the regression models. However, adjustment for pandemic phase accounts for temporal shifts in vaccination coverage and population immunity to some extent. Indigenous status was poorly recorded in the NCIMS database, with 65% of records being "not stated/unknown", meaning we were unable to adjust for this variable.

Case ascertainment is thought to have declined during the Omicron pandemic phase as a result of the PCR testing system struggling to cope with the far higher infection rate – at least until RATs became widely available. However, the non-significant interaction term between IRSD score and pandemic phase in our alternative analysis suggests that variation in case ascertainment was not influential for our results.

Finally, the observational design precludes causal attribution of specific mechanisms of adverse COVID-19 disease consequences.

## Implications

Despite the above limitations, the findings underscore the importance of recognising and addressing systemic inequities in health and the social determinants of health in pandemic preparedness and response. This includes intensifying chronic disease prevention and management in disadvantaged communities, addressing direct and indirect healthcare access barriers (including those created through pandemic control measures) [13], and investing in equity infrastructures – for example, place-based services and community partnerships that can be rapidly activated during emergencies [22]. Aside from potentially reducing disparities in disease consequences once infected, these measures may also temper other pathways through which disadvantage can lead to higher risk of hospitalisation and death, i.e., differential exposure, differential susceptibility to infection, and differential effectiveness of control measures [6].

The study also highlights the need for good-quality surveillance data, in particular, accurate and consistent recording of Indigenous status, and the ability to link them with clinical and vaccination data at metropolitan, state or national scale.

## Conclusions

This study found that individuals residing in more socioeconomically disadvantaged areas of Greater Sydney were more likely to be hospitalised and more likely to die due to COVID-19 once infected with SARS-CoV-2, within the context of a health system providing public hospital care. Reducing inequities in the burden of future health emergencies will require

measures that address differentials in exposure, susceptibility to infection, disease severity, and effectiveness of control measures across socioeconomic strata.

## Author contributions

**Conceptualization:** Christopher Standen, Esther Tordjmann.

**Data curation:** Christopher Standen.

**Formal analysis:** Christopher Standen.

**Funding acquisition:** Fiona Haigh.

**Investigation:** Christopher Standen.

**Methodology:** Christopher Standen.

**Project administration:** Christopher Standen, Fiona Haigh.

**Supervision:** Fiona Haigh.

**Visualization:** Christopher Standen.

**Writing – original draft:** Christopher Standen.

**Writing – review & editing:** Christopher Standen, Esther Tordjmann, James Wood, Fiona Haigh.

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
