## [Decision Letter · Decision Letter 0]

19 Dec 2025

Dear Dr. Standen,

plosone@plos.org. . . . • A letter that responds to each point raised by the academic editor and reviewer(s). You should upload this letter as a separate file labeled 'Response to Reviewers'.

We look forward to receiving your revised manuscript.

Kind regards,

Eric HY Lau, Ph.D.

Academic Editor

PLOS One

Journal Requirements:

“This work was funded by Sydney Local Health District through a memorandum of understanding with the University of New South Wales.”

7. We note that Figure 1, 2, 3 and 4 in your submission contain map images which may be copyrighted. All PLOS content is published under the Creative Commons Attribution License (CC BY 4.0), which means that the manuscript, images, and Supporting Information files will be freely available online, and any third party is permitted to access, download, copy, distribute, and use these materials in any way, even commercially, with proper attribution. For these reasons, we cannot publish previously copyrighted maps or satellite images created using proprietary data, such as Google software (Google Maps, Street View, and Earth). For more information, see our copyright guidelines: http://journals.plos.org/plosone/s/licenses-and-copyright.

1. You may seek permission from the original copyright holder of Figure 1, 2, 3 and 4 to publish the content specifically under the CC BY 4.0 license.

Additional Editor Comments:

The Authors are expected to address all comments by the Reviewer. In particular, please provide more details on the IRSD score, geographic scale and COVID-19 case definition, and consider temporal sub-analysis by stratifying the results by year (or by key epidemic period). In addition to the above comments, please address,

1. Table 2. What is the range for the IRSD score? The logistic regression estimated the effect of 1 unit IRSD increase but that could be a very small difference. Please consider rescaling the IRSD score in the model to improve interpretation.

2. Table 2. The authors may consider using 20-49 years as reference group, as COVID-19 deaths were rare among 0-19 years which may lead to unstable estimates.

Reviewer's Responses to Questions

**Comments to the Author**

1. Is the manuscript technically sound, and do the data support the conclusions?

Reviewer #1: Yes

2. Has the statistical analysis been performed appropriately and rigorously?

Reviewer #1: Yes

3. Have the authors made all data underlying the findings in their manuscript fully available?

Reviewer #1: Yes

4. Is the manuscript presented in an intelligible fashion and written in standard English?

Reviewer #1: Yes

Reviewer #1: Please see full reviewer report - Reccomendation: Accept with major revisions.

Summary:

Thank you for this submission - This paper investigates the relationship between area‑level socioeconomic disadvantage and COVID‑19 outcomes in Greater Sydney, Australia, focusing specifically on disease severity once infected. Using a retrospective cohort design, the authors analysed over 780,000 confirmed and probable cases between March 2020 and February 2022. They measured disadvantage via the Index of Relative Socio‑Economic Disadvantage (IRSD) and applied logistic regression models adjusted for age and gender. The findings show that lower IRSD scores (greater disadvantage) were consistently associated with higher odds of hospitalization and death, even in a setting with universal public hospital care. The authors conclude that inequities in comorbidity burden and possibly differences in care access underlie these disparities, and they recommend prioritizing chronic disease prevention and management in disadvantaged communities.

There has been similar work (Guajardo, et al. 2025) conducted examining how socioeconomic and demographic factors influenced COVID-19 outcomes (testing, infection, hospitalisation and deaths) in the Sydney Local Health District (SLHD). Key difference between these two works is the focus on the impact of public health restrictions (strict vs relaxed public health restrictions) on these outcomes with regards to socioeconomic and demographic factors and that this study focuses on a greater geographical area (beyond Sydney city) and larger time-period.

Placed in the context of existing literature, this study reinforces global evidence that socioeconomic disadvantage amplifies vulnerability to severe COVID‑19 outcomes. While similar associations have been documented nationally (Flavel, et al. 2022) and in Australian states (Queensland – Ward, et al. 2023; Victoria – Roder, et al. 2022), the originality of this work lies in its large-scale, population-level analysis, highlighting that inequities persist even when financial barriers to care are minimized (in the context of universal free health care in Australia). Thus, the paper is more confirmatory than novel, but it provides important local evidence that strengthens the broader narrative of social determinants of health in pandemic outcomes.

.

Reviewer #1: **Yes:** Selina WardSelina WardSelina WardSelina Ward

---

## [Author Response · Author response to Decision Letter 1]

25 Feb 2026

Methods

(lines 112-114) The methods section should specify how the IRSD is calculated and the geographic scale (e.g., SA1, SA2) that this is applied at. We have added a sentence and reference explaining how IRSD scores are calculated: “An IRSD score for a postal area is a weighted composite measure of 16 economic and social disadvantage indicators for that postal area, including the percentage of people aged 15 years and over who have no educational attainment, and the percentage of people who are unemployed [16].”

Methods

(lines 95-100) It is unclear whether self-reported rapid antigen tests (RATs) were included, which could affect case ascertainment. We have clarified in the Methods section that self-reported RAT results were included:

“From 12 January 2022, self-reporting of positive COVID-19 rapid antigen tests (RATs) was mandatory and could be done via the ‘Service NSW’ app or website, or by phone.”

We have also provided the proportion of self-reported positive cases in the Results section and addressed variation in case ascertainment in the Discussion.

Results

Figures 1-4 The current colour scale is difficult to interpret and should be revised. Suggest using multiple colours (for example - red, orange, green). Red, orange and green are difficult to distinguish for many people with colour vision deficiency. A monochromatic colour scheme is recommended by accessible publishing guidelines, such as WCAG. A monochromatic colour scheme works well when printed in greyscale.

Results This is acknowledged a limitation in the discussion – However, stratifying results by year (2020/2021/2022) could reveal shifts in associations. Thank you for this suggestion. We have addressed temporal heterogeneity by extending the analysis to include adjustment for pandemic phase, rather than calendar year, to capture major shifts in dominant SARS-CoV-2 variants, population immunity, and other factors.

Specifically, we classified cases into three pandemic phases based on diagnosis date: Ancestral (March 2020 to June 2021), Delta (June to November 2021), and Omicron (December 2021 to February 2022). Pandemic phase was included as a categorical covariate in the logistic regression models, which substantially improved overall model fit for both hospitalisation and death.

To assess whether socioeconomic gradients differed over time, we fitted pooled models including interaction terms between area-level socioeconomic disadvantage (IRSD) and pandemic phase. These interaction terms were not statistically significant for either hospitalisation or death, indicating no evidence that the association between socioeconomic disadvantage and disease severity varied across the three pandemic phases. Accordingly, final models are presented without the interaction terms.

The Abstract, Methods, Results, and Discussion sections have been updated to reflect these additional analyses and implications.

Methods - Figure 1 Zoomed in call out box of city and inner Sydney suburbs would provide better detail on heterogeneity of IRSD Adding additional inset maps for the inner city and other densely populated areas with small postal areas would make the map very cluttered. However, we agree it would be helpful to provide more detail on the heterogeneity of the IRSD variable. We have therefore provided a measure of the heterogeneity in the text:

“IRSD scores are standardised to have a mean of 1,000 and a standard deviation of 100 across Australia. In the study area, IRSD scores ranged from 711 to 1,122, with a mean of 1,029 and a standard deviation of 78.”

Results An epidemic curve would contextualize the temporal dynamics of cases We have added an epidemic curve with pandemic phases (Ancestral, Delta, Omicron) indicated to contextualise the temporal dynamics of COVID-19 cases over the study period (Fig 2).

Introduction

(lines 51-61) These two paragraphs can be difficult to follow for those unfamiliar with the pathways described by Katikireddi, et al. 2021 - Suggest rewording to clarify and improve flow of text. We have reworded this section of the Introduction to describe the Katikireddi et al. framework using more accessible language.

Methods

(lines 93-94) Suggest specifying what percent of the population (including eligible age groups) to add clarity. We have revised the Methods section to specify the percent of the eligible population aged ≥16 years that had received two vaccination doses.

Methods

(line 97) Suggest including mention of the private health system in Australia. We have clarified that PCR testing was provided by both public and private providers, and that some hospitalisations for COVID-19 were managed in private hospitals.

Methods

(lines 101-103) It is unclear why this sentence is included in the methods. We have moved discussion of infection-control measures in aged care settings to the Discussion, as it is of relevance to the observed number of COVID-19 hospitalisations and fatalities in older adults.

Results How many cases did not include a valid postal area and/or were acquired overseas. We have clarified that 2,527 cases were excluded because they were acquired overseas. All cases had a valid postal area after linking them to IRSD scores.

Results

Figures 2-4 Suggest rounding decimal places of legend to improve readability. We have reduced the number of decimal places to one.

Discussion

(lines 223-226) Suggest including reference to support these claims. In our original submission, the references were provided in the subsequent paragraph. We have merged the two paragraphs to make it clearer that those references are associated with these claims.

Discussion Suggest adding a brief overview of vaccination uptake to provide context. We have added the date COVID-19 vaccination became available, as well as coverage at the ends of the Delta and Omicron pandemic phases, to the Methods section.

Supplementary information Suggest including overview of demographics of cases included in analysis compared to local population. We have provided demographics of cases included in the analysis compared to the local population in Table 1.

Table 2 What is the range for the IRSD score? The logistic regression estimated the effect of 1 unit IRSD increase but that could be a very small difference. Please consider rescaling the IRSD score in the model to improve interpretation. Thank you for the suggestion. We have rescaled the IRSD variable to 100-point intervals and updated the Methods section to provide more background.

Table 2 The authors may consider using 20-49 years as reference group, as COVID-19 deaths were rare among 0-19 years which may lead to unstable estimates. We have changed the reference category for the age variable in both logistic regression models to be 20–49 years.

---

## [Decision Letter · Decision Letter 1]

22 Mar 2026

Associations between area-level socioeconomic disadvantage and COVID-19 disease consequences in Sydney, Australia: A retrospective cohort analysis

PONE-D-25-51852R1

Dear Dr. Standen,

We’re pleased to inform you that your manuscript has been judged scientifically suitable for publication and will be formally accepted for publication once it meets all outstanding technical requirements.

Kind regards,

Eric HY Lau, Ph.D.

Academic Editor

PLOS One

Additional Editor Comments (optional):

Thanks for addressing all the editor’s and reviewers' comments. Congratulations on the excellent work!

Reviewers' comments:

Reviewer's Responses to Questions

**Comments to the Author**

Reviewer #1: All comments have been addressed

2. Is the manuscript technically sound, and do the data support the conclusions?

Reviewer #1: Yes

3. Has the statistical analysis been performed appropriately and rigorously?

Reviewer #1: Yes

4. Have the authors made all data underlying the findings in their manuscript fully available?

Reviewer #1: Yes

5. Is the manuscript presented in an intelligible fashion and written in standard English?

Reviewer #1: Yes

Reviewer #1: The revisions have strengthened the manuscript, particularly in clarifying methodological details and contextualizing the findings within broader public health implications. Overall, the manuscript is suitable for publication.

.

Reviewer #1: **Yes:** Selina WardSelina WardSelina WardSelina Ward

---

## [Editor Report · Acceptance letter]

PONE-D-25-51852R1

PLOS One

Dear Dr. Standen,

I'm pleased to inform you that your manuscript has been deemed suitable for publication in PLOS One. Congratulations! Your manuscript is now being handed over to our production team.

Kind regards,

on behalf of

Dr. Eric HY Lau

Academic Editor

PLOS One